# Prediction of Spasticity through Upper Limb Active Range of Motion in Stroke Survivors: A Generalized Estimating Equation Model

**DOI:** 10.3390/bioengineering10111273

**Published:** 2023-11-01

**Authors:** Muhammad Adeel, Chih-Wei Peng, I-Jung Lee, Bor-Shing Lin

**Affiliations:** 1The Master Program in Smart Healthcare Management, International College of Sustainability Innovations, National Taipei University, New Taipei City 237303, Taiwan; dr.adeel215@gm.ntpu.edu.tw; 2School of Biomedical Engineering, College of Biomedical Engineering, Taipei Medical University, Taipei 11031, Taiwan; cwpeng@tmu.edu.tw; 3School of Gerontology and Long-Term Care, College of Nursing, Taipei Medical University, Taipei 11031, Taiwan; 4College of Electrical Engineering and Computer Science, National Taipei University, New Taipei City 237303, Taiwan; ijunglee0215@gmail.com; 5Department of Computer Science and Information Engineering, National Taipei University, New Taipei City 237303, Taiwan

**Keywords:** range of motion, limitation, assessment, chronic stroke, upper limb

## Abstract

Background: We aim to study the association between spasticity and active range of motion (ROM) during four repetitive functional tasks such as cone stacking (CS), fast flexion–extension (FFE), fast ball squeezing (FBS), and slow ball squeezing (SBS), and predicted spasticity models. Methods: An experimental study with control and stroke groups was conducted in a Medical Center. A total of sixty-four participants, including healthy control (*n* = 22; average age (years) = 54.68 ± 9.63; male/female = 12/10) and chronic stroke survivors (*n* = 42; average age = 56.83 ± 11.74; male/female = 32/10) were recruited. We employed a previously developed smart glove device mounted with multiple inertial measurement unit (IMU) sensors on the upper limbs of healthy and chronic stroke individuals. The recorded ROMs were used to predict subjective spasticity through generalized estimating equations (GEE) for the affected side. Results: The models have significant (*p* ≤ 0.05 *) prediction of spasticity for the elbow, thumb, index, middle, ring, and little fingers. Overall, during SBS and FFE activities, the maximum number of upper limb joints attained the greater average ROMs. For large joints, the elbow during CS and the wrist during FFE have the highest average ROMs, but smaller joints and the wrist have covered the highest average ROMs during FFE, FBS, and SBS activities. Conclusions: Thus, it is concluded that CS can be used for spasticity assessment of the elbow, FFE for the wrist, and SBS, FFE, and FBS activities for the thumb and finger joints in chronic stroke survivors.

## 1. Introduction

Spasticity is generally present in the stroke population; upper limb spasticity is more common than lower limb [1]. It impairs patients’ range of motion (ROM), defined as a rotation around a joint, which is an important component of evaluation in a clinical population [2], to the point where it prevents activities of daily living (ADL) and functional recovery, which is detrimental to effective rehabilitation [3]. The ROM of the joint is affected by spasticity due to the changes in the muscle–tendon length [4] in a shortened position over time [5].

A study by Jeanette et al. [6] reported that after a stroke, early quantitative measurements of hand spasticity may be able to forecast functional recovery and direct targeted rehabilitation measures. The most widely used clinical tool is the modified Ashworth scale (MAS) because it is quite easy to employ [7]. Although it is conducive to utilizing the MAS, it has some shortcomings. Firstly, it is a subjective tool and has low interrater reliability [8]. Secondly, it entails a clinician conducting an assessment upon a follow-up visit and cannot be applied without an evaluator; therefore, an objective tool is required to assess spasticity.

For this purpose, different systems with inertial measurement units (IMUs) and surface electromyography (sEMG) sensors are being developed. One proposed by Ang et al. [9] used three IMUs to measure the upper limb joint angle, velocity, and acceleration during upper limb activity. They predicted the velocity-dependent tonic stretch reflex threshold and demonstrated a high correlation with the MAS score, which can be an indicator of spasticity. Using inertial sensors, Kim et al. [10] utilized a machine-learning approach to assessing spasticity during passive elbow stretch in stroke and spinal cord injury patients. They classified the degree of spastic movement and reported that the method was comparable to assigning a MAS score with 95.4% accuracy. Another study reported a satisfactory performance of regression models in terms of low mean square error (MSE: 0.06, 0.14, and 0.47) in assessing the spasticity of stroke patients with passive elbow movements using wearable sEMG and IMUs [11]. However, those studies assessed upper-limb spasticity through passive movements, which could not reveal how spasticity affects the patient’s activities. Therefore, some studies have assessed spasticity in stroke using active movements. The study by Bai et al. [12] designed a system to record the upper limb’s ROM before and after using botulinum; however, this study only focused on the quality of movement, not the spasticity. Another study by Chen et al. [13] employed elbow flexion and extension repetitive voluntary movements to assess elbow spasticity in stroke patients. They revealed that the random forest machine learning technique using IMUs and sEMGs had the greatest effect (F1-score = 0.95) compared with the sEMG signal (F1-score = 0.76) or motion signal only (F1-score = 0.71). Lin el al. proposed a multi-sensor system with IMUs to assess finger joint spasticity by performing cone stacking (CS), slow flexion–extension (SFE), fast flexion–extension (FFE), slow ball squeezing (SBS), and fast ball squeezing (FBS) [14]. The comparison with the previous studies is presented in Table 1.

Although the relations between spasticity and voluntary movements have been discussed in the above studies, they require lots of feature engineering on the raw data to build a model for spasticity, which is a time-consuming process. Furthermore, it is difficult for the therapists to know the relationship between voluntary movements and spasticity because the sensor data are not easily understood. Owing to the spasticity effect on ROM, the ROM of the stroke patients’ upper limb joint while performing voluntary movements should be used to determine the spasticity level instead of using sensor data. Moreover, most of the research has built the models with the data from both the left- and right-affected sides, which might have different motion characteristics. Haaland et al. [16] reported that righthanded patients with left-hemisphere lesions showed notable deficits in movement speed, while right-hemisphere lesions displayed significant final position errors [17]. Such ipsilesional abnormalities have been linked to significantly worse performance on functional measures, such as simulated ADL [18,19,20].

Hence, to test our hypothesis for the AROM and MAS score correlation for the affected upper limb joints, and to solve the aforementioned limitation of using sensor data to assess spasticity in chronic stroke, we used a laboratory-developed wearable system and performed four voluntary tasks to record the AROM and to assess the spasticity level of the upper limb joints in stroke survivors, by wearing a smart-glove device on both upper limbs. We used generalized estimating equation (GEE) models to avoid time-consuming processes like dimension reduction and feature selection techniques on sensor data. Hence, the main goal of our study was to predict the spasticity using AROM following the four specified tasks in stroke survivors and computed models for both the right and left sides. Despite the limitations of the MAS scale, in our current study, we used this scale to obtain the subjective MAS grades and recorded the AROM during four voluntary tasks. Through this research, we aimed to highlight a robust and convenient method for the evaluation of spasticity and the limitation of AROM in the clinical population.

## 2. Materials and Methods

### 2.1. Wearable System and Joint Angle Measurement

The data were collected through an upper limb motion capture device (UMCD) comprised of a sensory glove and motion tracking device for the upper arm (MTD-UA), in which a total of 19 IMUs were mounted. The hardware and software design with the device calibration procedure and sensitivity were presented in previous studies [14,21]. The data from the UMCD were used to calculate the joint angle of 16 upper limb joints, including the elbow, wrist, thumb (first metacarpophalangeal, MP1; and interphalangeal, IP), finger 2 (index metacarpophalangeal, MP2; index proximal interphalangeal, PIP2; and index distal interphalangeal, DIP2), finger 3 (middle finger: MP3, PIP3, and DIP3), finger 4 (ring finger: MP4, PIP4, and DIP4), and finger 5 (little finger: MP5, PIP5, and DIP5), respectively. The C# programming software converted the data received from the UMCD into acceleration, angular velocity, and magnetic field strength. The sensor fusion algorithm proposed by Madgwick et al. [22] used the quaternion of the attitude at the current time for every two adjacent IMUs to calculate the joint angle between the two adjacent IMUs [10,11] (Figure 1).

### 2.2. Data Acquisition Protocol

The study recruited 22 healthy and 42 stroke participants through convenience sampling (Figure 2) and was conducted in the Medical Center of a hospital. The study procedures followed the Declaration of Helsinki: Ethical Principles for Medical Research Involving Human Subjects (version October 2013) and were approved by the Institutional Review Board (IRB) (code: 11002-007). The participants were included if they were (1) aged 20–80 years and (2) were able to sit on a chair for about 40 min; they were excluded if they had (1) symptoms of unilateral neglect or attention deficit, (2) cognitive or language deterioration and not be able to comprehend and execute the individual tasks, (3) upper-limb disability due to musculoskeletal or peripheral nervous system lesions before the onset of stroke, or (4) diagnosed with dementia or depression assessed by a rehabilitation doctor and a physical therapist. Each participant signed a written consent form explained by the experienced researcher.

An experienced physical therapist performed the baseline evaluation of the participants, and spasticity was assessed using a 6-point MAS scale with the following scores, 0 = no increase in muscle tone; 1 = slight increase in muscle tone, manifested by a catch and release or by minimal resistance at the end of the range of motion when the affected part is moved in flexion or extension; 1+ = slight increase in muscle tone, manifested by a catch, followed by minimal resistance throughout the remainder (less than half) of the ROM; 2 = more marked increase in muscle tone through most of the ROM, but the affected part easily moved; 3 = considerable increase in muscle tone, passive movement difficult; and 4 = affected part rigid in flexion or extension [7] for the affected-side elbow, wrist, MPs of the thumb, index, middle, ring, and little fingers. The MPs of the index, middle, ring, and little fingers were evaluated together to give only one spasticity level. Since the experiments required the patients to perform the specified tasks with their hands voluntarily, only the patients with a MAS of 0 to 2 were recruited in this study. The stroke patients’ spasticity was categorized according to the MAS into three subclasses: healthy/no = 0, mild = 1, and moderate = +1 and 2, respectively. One researcher applied the UMCD on the subject’s upper limb and confirmed the correct alignment of the IMUs on the fingers, hand, and upper arm. The researcher then demonstrated to the healthy and stroke participants how to perform four tasks, including (1) CS, (2) FFE, (3) FBS, and (4) SBS, using both upper limbs. The four tasks were chosen because they are the most frequently used in occupational therapy and most stroke patients get familiar with them to perform more readily [14].

Each participant performed the task at their preferred speed with a resting period between each task; however, the FFE and FBS tasks were performed at higher speeds than the preferred pace. The participants put the test hand on a table that was 75–80 cm high before each task. The table’s legs were secured to the floor and its height was adjusted according to the participant’s height to avoid any unforeseen movement. For the CS task, two cone bases were arranged 20 cm from the edge of the table. Before the task, ten cones were placed near the side opposite to the participant’s test side, that is, if the left side were to be tested, the ten cones were placed on the cone base on the right side and vice versa. The participants transferred the ten cones from one cone base to the other as rapidly as they could after getting the signal to begin the test. The participants were not allowed to lean their trunks to any side during the test. After moving all the cones, the participant returned the test hand to the initial position. They performed 50 rapid flexions and extensions for the FFE and squeezed a ball rapidly and slowly 50 times for the FBS and SBS tasks. The voluntary tasks are explained in more detail in this study [14].

### 2.3. Model for Spasticity Assessment

The time series data of each joint collected with a sampling rate of 50 Hz [21] was processed as an average of 10 repetitions for CS and 20 repetitions for FFE, FBS, and SBS in Matlab (R2021a, MathWorks, Inc., Natick, MA, USA). A total of 20 variables were processed for the GEE models: 16 for the AROMs of the upper limb joints as independent variables and 4 for the MAS spasticity for the elbow, wrist, thumb, and finger as dependent variables, respectively.

A statistical package for social sciences (SPSS) was used for further analysis and model prediction. The normality of the data was checked through a bell-shaped histogram, which represented a normal distribution. A GEE [23] method was used to predict spasticity by utilizing a backward deletion approach for the repeated measured data of four activities’ AROMs. The GEE expands the generalized linear models [24,25], which comprise simple linear regression [23,26].

Two types of models were computed, namely, the model for the right-dominant right-affected side and the right-dominant left-affected side. Each activity’s ROM was correlated with each joint MAS score in the GEE model to predict the spasticity. The working correlation matrix chosen was first-order autoregressive (AR1), which means repeated measurements have a first-order autoregressive relationship in which immediately preceding values are used to predict the value at the present time (Figure 3).

The GEE models were evaluated through the estimate (β), standard error (SE), 95% confidence interval (CI), and *p*-value [27]. A valid measure has a smaller 95% CI, whereas a larger 95% CI means a less accurate model [28]. The GEE model fit was examined through quasi-likelihood under the independence model criteria (QIC), which denotes that the lower the QIC, the better the model fit [29]. The level of significance was set to *p* ≤ 0.05 *, *p* ≤ 0.001 **, and *p* ≤ 0.0001 ***. The AROMs of four tasks for healthy control, unaffected and affected sides in stroke survivors are presented through box and bar plots.

## 3. Results

The main characteristics of the participants showed that 12 out of 22 were males and the rest were females in the healthy group, while in the stroke group, 32 were male and 10 were female participants. The average age for both groups ranged from 54–57 years. Most of the participants were right-handed in both groups. Among the stroke participants, 19 were right-side affected, 22 were left-side affected, and 1 with both sides affected, as presented in Table 2.

### 3.1. GEE Models of Spasticity for the Affected Sides

Out of 42 stroke patients, 17 were right-dominant right-affected, 19 were right-dominant left-affected, 2 were left-dominant right-affected, 3 were left-dominant left-affected, and 1 had both sides affected. The GEE models were predicted only for the right-dominant affected patients due to the small sample size for the left-dominant affected patients.

#### 3.1.1. Right-Dominant Right-Affected Side Models (*n* = 17)

Table 3 shows that the GEE models significantly predicted spasticity for the right-dominant right-affected upper limb joints except for the elbow, wrist, and thumb (*p* = 0.001 **~0.0001 ***). For finger 2, finger 3, and finger 4, the ROMs of all joints predicted spasticity significantly during four tasks, and the QIC of the models were 22, 19, and 22, respectively. For finger 5, only the ROMs of PIP predicted spasticity significantly while performing FBS and SBS, and the QIC was 12.

#### 3.1.2. Right-Dominant Left-Affected Side Models (*n* = 19)

Table 4 presents the significant models, except for wrist and finger 5 (*p* = 0.001 **~0.0001 ***). The ROM of the elbow predicted spasticity significantly only during FBS, and the QIC of the model was 9. The ROM of the IP on the thumb predicted spasticity significantly only while performing FFE, and the QIC was 8. For finger 2, the ROMs of MP, PIP, and DIP on finger 2 predicted models while performing CS, FFE, FBS, and SBS, and the QIC was 15. For finger 3, the ROMs of PIP and DIP on finger 3 predicted spasticity while performing CS, FFE, FBS, and SBS, and the QIC was 14. For finger 4, the ROMs of MP, PIP, and DIP on finger 4 predicted spasticity significantly while performing FFE, FBS, and SBS, and the QIC was 11.

### 3.2. ROMs of Healthy Control and Stroke Survivors

The healthy participants in Figure 4a show that FBS covered the maximum range from 15 to 110 degrees. The medians for CS = 47 degrees, FFE = 65 degrees, FBS = 48 degrees, and SBS = 51 degrees, respectively. Regarding the stroke patients’ affected side, Figure 4b shows that FBS covered the maximum range from 3 to 126 degrees, and medians for CS = 36, FFE = 41, FBS, and SBS = 42, respectively. The unaffected side of stroke patients in Figure 4c shows that FFE covered the maximum range from 3 to 159 degrees, and medians for CS = 32, FFE = 47, FBS, and SBS = 43, respectively.

### 3.3. Average AROMs of the Upper Limb Joints in Stroke Patients

The right-affected side had large average ROMs for the elbow during CS and FFE, thumb during SBS, finger 2 during FFE, FBS, and SBS, finger 3 and finger 4 during FFE, and finger 5 during FBS, respectively. The left affected side had greater average ROMs for the elbow during SBS, wrist during all four activities, thumb during CS, FFE, and FBS, finger 2 during CS, finger 3 and finger 4 during CS, FBS, and SBS, and finger 5 during CS, and FFE, respectively (Figure 5). Table 5 and Table 6 show the involvement of the upper limb joints during the four voluntary tasks and a comparison between affected and unaffected for different categories of spasticity in chronic stroke.

## 4. Discussion

This is the first study to explore the relationship between spasticity through MAS score and AROM in chronic stroke survivors, and to classify three categories (healthy, mild, and moderate) of ROM for stroke participants’ unaffected and affected sides. Two different conditions, including the right-dominant right-affected and the right-dominant left-affected-sides’ GEE models, were built. According to our results, the AROM of upper limb joints can assist the subjective spasticity assessment (MAS) in stroke survivors. The stiffness or weakening of the affected side’s muscles contributes to the development of contractures, leading to increased muscle spasticity in stroke. In this way, the measured ROM of the upper limb joints during four functional activities was related to the degree of spasticity. As a result, this relationship between ROM and spasticity can help physicians and physical therapists with an objective measure in the early diagnosis of the extent of spasticity.

The GEE models for stroke survivors have good significance and prediction for finger 2, finger 3, and finger 4, for both the right and left affected sides. Although most of the participants were right-handed, the left-affected side models predicted spasticity with more upper limb joints than the right-affected side. The reason could be that more stroke survivors’ affected sides were left-based in our study. While both affected sides reached a significant level, there was a lower QIC and narrower CI for the left-affected side (QIC, 8~15) and (95% CI, 0.01~0.03) than the right-affected side (QIC, 4~22) and (95% CI, 0.07~0.10) respectively, makes it more predictable [23,27,28] (Table 3 and Table 4). A previous study confirmed that five functional tasks of the upper limb in stroke participants could provide a significant prediction of finger spasticity [14]. This is further confirmed in our current study, and hence, we obtained the significant GEE models by correlating the spasticity scores with the respective joints’ ROMs.

In comparison to the healthy control, stroke survivors on the affected side have a lesser ROM when assessed with a higher MAS for the elbow (unaffected, 54.53 (30.71) vs. affected, 51.26 (21.22), *p* = 0.738) and wrist (unaffected, 32.86 (9.51) vs. affected, 27.60 (10.71), *p* = 0.704) joints, but finger joints did not show this trend possibly due to the lack of individual joint spasticity assessment (Table 6). Because of the presence of spasticity on the affected side, the ROMs were limited, and the stroke survivors did not cover higher degrees during the four functional tasks. The results can infer that more upper-limb joints in left-affected stroke survivors’ have attained greater average ROMs than the right-affected stroke. This depicts that the right-affected stroke has more limitations in the upper-limb ROM as compared to left-affected stroke (Figure 4 and Table 5).

Based on the literature, it is difficult to state clearly that the GEE method can predict significant models for only the left-affected side but not for the right-affected side. This may be due to the smaller number of participants for the right-affected side, different levels of spasticity, and varying degrees of activity limitations on each side. Several studies [18,30] have demonstrated that deficits were more pronounced in left-hemisphere-damaged patients [30]. However, we are the first to predict the significant models for spasticity detection in chronic stroke using AROM activities for the right- and left-affected stroke survivors.

All the previous studies [10,11,13,15] presented in Table 1 used machine learning approaches to assess spasticity from elbow joint movements with a significant correlation (r = 0.93), but Lin et al. [14] and our current study have utilized elbow, wrist, and fingers voluntary ROM to assess spasticity in chronic stroke. In line with previously published research [14], which found a significant (*p* < 0.05 *) correlation (r = 0.94), we have computed significant GEE models (*p* < 0.05 *) for right- and left-affected sides for upper limb spasticity. Another study predicted the severity of spasticity of the elbow joint using tonic stretch reflex threshold (TSRT) from upper limb muscles’ EMG with a significant prediction (*p* = 0.0002 **~0.0003 **) [9]. While in our study, the significant models were predicted (*p* = 0.001 **~0.0001 ***) for upper limb joints. Hence, our study’s strength can be explained twofold: (1) the prediction of significant spasticity models for the right and the left-affected sides of stroke survivors and (2) the utilization of a common type of activities for spasticity assessment, which is the main component of the rehabilitation program in stroke survivors.

According to the definition of spasticity, the correlation between the limitation in the AROM during these four voluntary tasks and the TSRT, sEMG, or speed of movement is unclear. Even though we have tested slow and fast-speed tasks in stroke survivors and found out that fast-speed tasks (FFE and FBS) utilized more upper limb joints as compared to slow-speed tasks (CS and SBS) in Table 5, the CS and FFE involved the elbow and wrist joints while the FBS and SBS involved the wrist and finger joints due to their movement characteristics.

## 5. Conclusions

CS can be used for spasticity assessment of the elbow, FFE for the wrist, and SBS, FFE, and FBS activities for thumb and finger joints in chronic stroke survivors.

### 5.1. Limitations

(1) Reproducibility and sensitivity of spasticity models for stroke and healthy people were not tested; (2) spasticity models only computed for MAS grades 0–2 spasticity and MAS grade 3 was not included; (3) the sample size of both groups did not balance and control group has fewer participants than stroke group; (4) no muscle strength was recorded for the stroke survivors; (5) spasticity was assessed only once, it should be reassessed after four functional tasks; (6) did not include Brunnstrom recovery stage; and (7) did not collect sEMG and TSRT during four functional tasks.

### 5.2. Future Directions

Future research with a balanced sample size of both groups, calculated through G-power and randomized controlled design, should be conducted. The TSRT should be measured in future studies, along with sEMG activation of the upper limb muscles in stroke survivors. A future study will be required to test each task at different speeds for the exact involvement of each upper limb joint and muscle in chronic stroke survivors. It will also focus on explaining the fact that either left- or right-affected side models could be better to predict spasticity, or only one side, and to predict individual models for the healthy/no, mild, moderate, and severe spastic stroke survivors. 

### 5.3. Contributions

Firstly, this study found a relationship between AROM and spasticity through MAS scores with four voluntary activities, which had not been researched in previous studies. By using AROM, therapists can easily know the relationship between the patients’ movements and their spasticity level objectively in the early stage of stroke and design an appropriate rehabilitation program. Secondly, we built the GEE models for the right- and left-affected sides separately because the extent of spasticity is different for both affected sides and ensues the varying limitations in AROMs of the upper limb joints; this has also not been researched in previous studies.

## Figures and Tables

**Figure 1 bioengineering-10-01273-f001:**
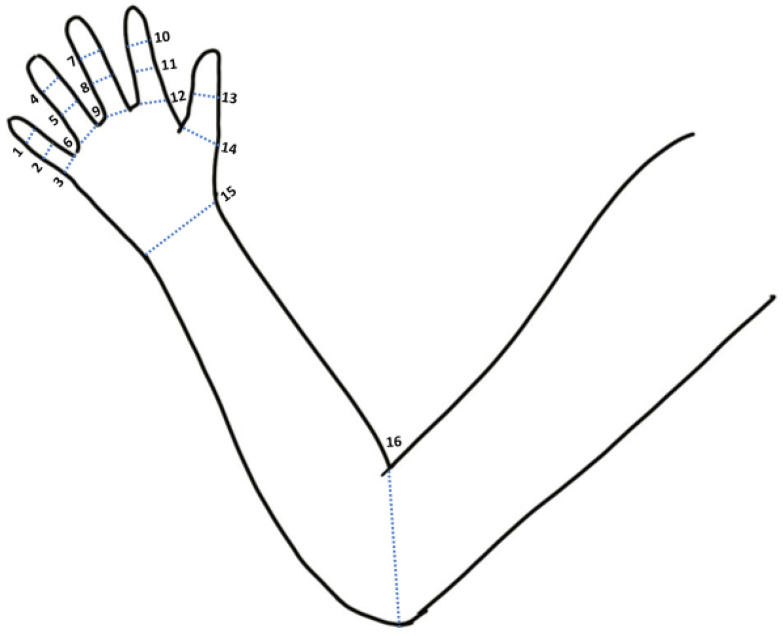
Joint angle of upper limb joints including 1, DIP5; 2, PIP5; 3, MP5; 4, DIP4; 5, PIP4; 6, MP4; 7, DIP3; 8, PIP3; 9, MP3; 10, DIP2; 11, PIP2; 12, MP2; 13, IP; 14, MP1; 15, wrist; and 16, elbow.

**Figure 2 bioengineering-10-01273-f002:**
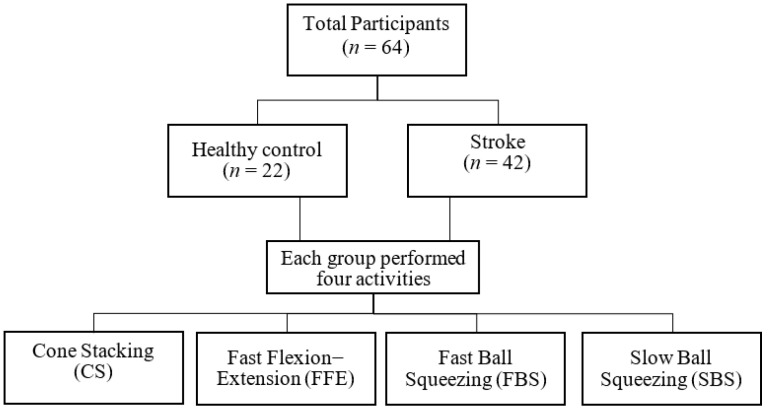
Subject recruitment and four specified activities.

**Figure 3 bioengineering-10-01273-f003:**
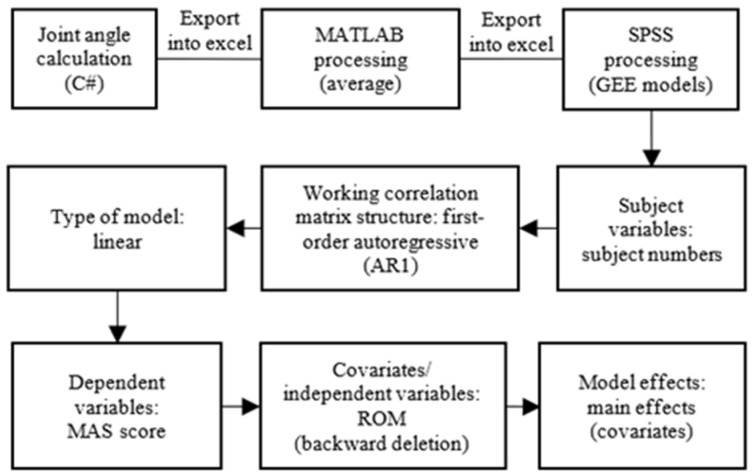
Flow chart for data processing and GEE model computation.

**Figure 4 bioengineering-10-01273-f004:**
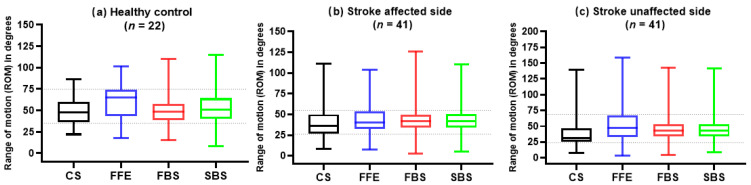
Active ROMs of four functional tasks for healthy and stroke participants: (**a**) healthy control, (**b**) stroke-affected side, and (**c**) stroke-unaffected side.

**Figure 5 bioengineering-10-01273-f005:**
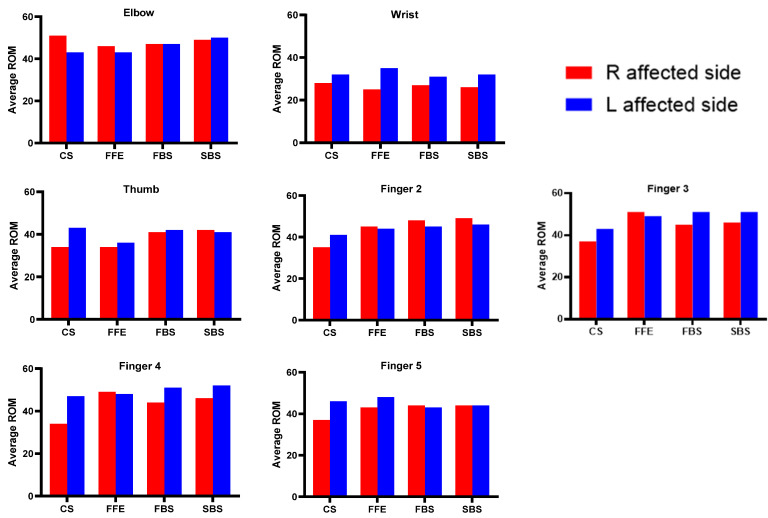
The right- and left-affected side average ROMs of stroke survivors during four activities.

**Table 1 bioengineering-10-01273-t001:** Comparison with the previous studies.

	Our Study	Lin et al. [14]	Park et al. [15]	Zhang et al. [11]	Kim et al. [10]	Chen et al. [13]
Type of tasks	V	V	P	P	P	V
Assessment scale	MAS	MAS	MAS	MAS	MAS	MAS
Included joints	E, W, T, F	E, W, T, F	E	E	E	E
Analysis of right/left-affected side	Yes	No	No	No	No	No
Analysis of each finger joint	Yes	No	No	No	No	No
Correlation (*r*)	N/A	E: 0.93 W: 0.94 T: 0.92 F: 0.92	0.83	0.93	N/A	N/A
*p*-value	<0.05 *	<0.05 *	N/A	<0.05 *	N/A	<0.05 *

V: voluntary, P: passive, E: elbow, W: wrist, T: thumb, F: fingers, *: represents the significant *p*-value.

**Table 2 bioengineering-10-01273-t002:** Study demographics (*n* = 64).

	Healthy Control (*n* = 22)	Stroke (*n* = 42)
Gender (male/female)	12/10	32/10
Age (years)	54.68 ± 9.63	56.83 ± 11.74
Dominant side (right/left)	21/1	37/5
Affected side (right/left/both)		19/22/1
Type of injury (hemorrhagic/ischemic)		11/31
Time since stroke (months)		38.52 ± 48.22
MAS elbow (healthy/mild/moderate)		7/15/20
MAS wrist (healthy/mild/moderate)		19/9/14
MAS thumb (healthy/mild/moderate)		23/16/3
MAS finger (healthy/mild/moderate)		20/13/9

**Table 3 bioengineering-10-01273-t003:** GEE models for spasticity (MAS) in stroke survivors (right-dominant right-affected side) (*n* = 17/42).

Finger	Variable	Estimate (β)	SE	95% CI (Lower~Upper)	*p*
Finger 2 (QIC = 22)	Intercept	12.76	0.44	11.89~13.63	0.0001 ***
	CS_DIP2	−0.04	0.00	−0.05~−0.03	0.0001 ***
	CS_PIP2	−0.02	0.00	−0.02~−0.01	0.0001 ***
	CS_MP2	0.39	0.01	0.37~0.41	0.0001 ***
	FFE_DIP2	−0.26	0.01	−0.27~−0.24	0.0001 ***
	FFE_PIP2	0.01	0.00	0.01~0.02	0.002 *
	FFE_MP2	−0.09	0.00	−0.09~−0.08	0.0001 ***
	FBS_DIP2	−0.08	0.01	−0.09~−0.06	0.0001 ***
	FBS_PIP2	−0.08	0.01	−0.10~−0.06	0.0001 ***
	FBS_MP2	0.06	0.01	0.05~0.07	0.0001 ***
	SBS_DIP2	0.20	0.01	0.19~0.22	0.0001 ***
	SBS_PIP2	−0.03	0.01	−0.05~−0.01	0.001 **
	SBS_MP2	−0.26	0.01	−0.28~−0.23	0.0001 ***
Finger 3 (QIC = 19)	Intercept	−1.97	1.73	−5.36~1.42	0.255
	CS_DIP3	−0.09	0.01	−0.11~−0.06	0.0001 ***
	CS_PIP3	0.04	0.01	0.02~0.06	0.0001 ***
	FFE_PIP3	0.05	0.01	0.03~0.06	0.0001 ***
	FFE_MP3	0.07	0.01	0.05~0.09	0.0001 ***
	FBS_DIP3	−0.08	0.03	−0.14~−0.02	0.007 *
	FBS_PIP3	-0.16	0.03	−0.21~−0.10	0.0001 ***
	FBS_MP3	−0.09	0.01	−0.11~−0.08	0.0001 ***
	SBS_DIP3	0.08	0.03	0.03~0.14	0.004 *
	SBS_PIP3	0.18	0.03	0.12~0.23	0.0001 ***
Finger 4 (QIC = 22)	Intercept	−1.59	0.41	−2.39~−0.80	0.0001 ***
	CS_DIP4	−0.05	0.01	−0.06~−0.04	0.0001 ***
	CS_PIP4	−0.02	0.00	−0.02~−0.01	0.0001 ***
	CS_MP4	−0.02	0.01	−0.03~−0.00	0.026 *
	FFE_DIP4	−0.02	0.00	−0.02~−0.01	0.0001 ***
	FFE_PIP4	0.03	0.00	0.02~0.03	0.0001 ***
	FFE_MP4	0.08	0.01	0.06~0.10	0.0001 ***
	FBS_DIP4	−0.06	0.01	−0.07~−0.05	0.0001 ***
	FBS_MP4	−0.11	0.01	−0.12~−0.09	0.0001 ***
	SBS_DIP4	0.08	0.01	0.07~0.09	0.0001 ***
	SBS_PIP4	0.03	0.01	0.02~0.04	0.0001 ***
	SBS_MP4	0.05	0.01	0.03~0.06	0.0001 ***
Finger 5 (QIC = 12)	Intercept	−0.40	0.49	−1.37~0.56	0.414
	FBS_PIP5	−0.02	0.01	−0.03~−0.01	0.0001 ***
	SBS_PIP5	0.04	0.01	0.02~0.06	0.0001 ***

QIC, quasi-likelihood under independence model Criterion; β, estimate coefficient; SE, standard error; CI, confidence interval; CS, cone stacking; FFE, fast flexion–extension; FBS, fast ball squeezing; SBS, slow ball squeezing; IP, interphalangeal joint; DIP, distal interphalangeal joint; PIP, proximal interphalangeal joint; MP, metacarpophalangeal joint. The level of significance was set to *p* ≤ 0.05 *, *p* ≤ 0.001 **, and *p* ≤ 0.0001 ***.

**Table 4 bioengineering-10-01273-t004:** GEE models for spasticity (MAS) in stroke survivors (right-dominant left-affected side) (*n* = 19/42).

Joint	Variable	Estimate (β)	SE	95% CI (Lower~Upper)	*p*
Elbow (QIC 9)	Intercept	0.26	0.30	−0.33~0.86	0.387
	FBS	0.02	0.01	0.01~0.03	0.0001 ***
Thumb (QIC 8)	Intercept	−0.17	0.16	−0.49~0.14	0.284
	FFE_IP	0.01	0.00	0.01~0.02	0.0001 ***
Finger 2 (QIC 15)	Intercept	−1.65	0.42	−2.47~−0.83	0.0001 ***
	CS_DIP2	−0.02	0.01	−0.03~−0.00	0.023 *
	FFE_DIP2	0.01	0.03	0.01~0.02	0.0001 ***
	FFE_MP2	0.05	0.01	0.04~0.06	0.0001 ***
	FBS_MP2	−0.04	0.01	−0.05~−0.02	0.0001 ***
	SBS_PIP2	0.02	0.01	0.00~0.03	0.030 *
	SBS_MP2	0.03	0.01	0.01~0.05	0.016 *
Finger 3 (QIC 14)	Intercept	−1.20	0.33	−1.84~−0.56	0.0001 ***
	CS_DIP3	0.01	0.00	0.00~0.01	0.037 *
	CS_PIP3	0.02	0.01	0.01~0.04	0.0001 ***
	FFE_DIP3	0.01	0.00	0.00~0.02	0.004 *
	FBS_DIP3	−0.01	0.0	−0.02~−0.00	0.002 *
	FBS_PIP3	0.03	0.01	0.02~0.05	0.0001 ***
	SBS_PIP3	−0.03	0.01	−0.04~−0.01	0.0001 ***
Finger 4 (QIC 11)	Intercept	−1.25	0.34	−1.92~−0.59	0.0001 ***
	FFE_PIP4	0.01	0.00	0.00~0.02	0.001 **
	FFE_MP4	0.03	0.01	0.01~0.04	0.0001 ***
	FBS_DIP4	0.02	0.05	0.01~0.02	0.002 *
	FBS_MP4	−0.01	0.00	−0.02~−0.01	0.0001 ***
	SBS_DIP4	−0.02	0.01	−0.03~−0.01	0.0001 ***
	SBS_PIP4	0.01	0.00	0.01~0.01	0.0001 ***

QIC, quasi-likelihood under independence model Criterion; β, estimate coefficient; SE, standard error; CI, confidence interval; CS, cone stacking; FFE, fast flexion–extension; FBS, fast ball squeezing; SBS, slow ball squeezing; IP, interphalangeal joint; DIP, distal interphalangeal joint; PIP, proximal interphalangeal joint; MP, metacarpophalangeal joint. The level of significance was set to *p* ≤ 0.05 *, *p* ≤ 0.001 **, and *p* ≤ 0.0001 ***.

**Table 5 bioengineering-10-01273-t005:** The involvement of the upper limb joints during four activities in stroke patients (*n* = 36).

CS	FFE	FBS	SBS
Elbow_R	Elbow_R	MP1_R	IP_R
Wrist_L	MP3_R	PIP2_R	MP2_R
MP1_L	PIP3_R	MP5_R	DIP2_R
DIP3_L	DIP3_R	PIP5_R	PIP3_R
PIP4_L	MP4_R	Wrist_L	PIP4_R
MP5_L	DIP4_R	IP_L	PIP5_R
	DIP5_R	PIP2_L	Elbow_L
	Wrist_L	MP3_L	Wrist_L
	DIP2_L	DIP4_L	MP2_L
	PIP5_L		PIP2_L
	DIP5_L		PIP3_L
			MP4_L

Out of 42 patients, 6 were not included because 2 had left-dominant right-affected side and 3 had left-dominant left-affected side, while 1 had both sides affected. CS, cone stacking; FFE, fast flexion–extension; FBS, fast ball squeezing; SBS, slow ball squeezing; IP, thumb interphalangeal joint; DIP 2~5, finger 2~5 distal interphalangeal joint; PIP 2~5, finger 2~5 proximal interphalangeal joint; MP 1~5, finger 1~5 metacarpophalangeal joints; R, right-affected side; L, left-affected side.

**Table 6 bioengineering-10-01273-t006:** Comparison of the active range of motion (AROM) based on the spasticity levels in chronic stroke survivors (*n* = 41).

Joints	Spasticity	MAS_0		MAS_1		MAS_2	
		Mean (SD)	*p*-Value	Mean (SD)	*p*-Value	Mean (SD)	*p*-Value
Elbow	Un_affected	45.74 (30.98)	0.450	43.32 (31.94)	0.915	54.53 (30.71)	0.738
	Affected	31.93 (17.27)		44.36 (16.98)		51.26 (21.22)	
Wrist	Un_affected	40.23 (25.13)	0.052	49.55 (29.26)	0.185	32.86 (9.51)	0.704
	Affected	29.81 (12.29)		32.39 (15.11)		27.60 (10.71)	
Thumb	Un_affected	35.53 (13.00)	0.235	36.09 (7.48)	0.330	34.88 (8.39)	0.241
	Affected	37.88 (10.51)		39.20 (9.95)		61.73 (23.79)	
Finger_2	Un_affected	46.44 (9.82)	0.126	50.16 (10.45)	0.294	41.79 (5.36)	0.032
	Affected	42.09 (8.88)		46.87 (9.35)		48.12 (7.17)	
Finger_3	Un_affected	50.29 (12.72)	0.079	56.41 (14.81)	0.270	46.00 (6.13)	0.237
	Affected	44.89 (7.70)		50.74 (16.70)		49.34 (10.88)	
Finger_4	Un_affected	48.49 (11.38)	0.087	57.32 (18.08)	0.245	48.42 (10.58)	0.663
	Affected	43.48 (7.05)		49.60 (14.13)		50.92 (11.96)	
Finger_5	Un_affected	43.40 (9.71)	0.909	51.76 (15.39)	0.416	43.04 (9.19)	0.389
	Affected	43.11 (10.98)		47.45 (14.86)		46.15 (8.48)	

MAS, modified Ashworth scale; MAS_0, no spasticity; MAS_1, mild spasticity; MAS_2, moderate spasticity; pairwise *t*-test between unaffected and affected sides are used. Out of 42 participants, 1 was excluded because of both affected sides.

## Data Availability

Not applicable.

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
