# Peer review of "Prediction of Spasticity through Upper Limb Active Range of Motion in Stroke Survivors: A Generalized Estimating Equation Model"

_bioengineering, 2023, doi:10.3390/bioengineering10111273_

Round 1

Reviewer 1 Report

Comments and Suggestions for Authors

1. Shoulder is very important for Stroke recovery. Comment on impact of shoulder on recovery

2. Stages of recovery can affect spasticity. Were all groups at same level of recovery?

3. What medications were patients taking?

4. What was the level of fitness pre stroke?

5. Impact of smoking and recovery

6. Presence or absence of partner influencing recovery?

Comments on the Quality of English Language

English ok

Author Response

Author’s replies to the comments of Reviewer 1:

[Comment 1]

The shoulder is very important for Stroke recovery. Comment on impact of shoulder on recovery

Answer:

Many thanks for the reviewer’s comments. We agree with the comment that the shoulder is very important for Stroke recovery and completes the kinetic and kinematic chain of the upper extremity in the execution of everyday movements. However, we did not include the shoulder in our current study because of the device design and we tested it on the elbow, wrist, and finger joints only. But in the future, studies should be conducted to test the device and predict models on the Stroke shoulder’s recovery as well which is an important direction for future research.

[Comment 2]

Stages of recovery can affect spasticity. Were all groups at same level of recovery?

Answer:

Many thanks for the reviewer’s comments. We agree with the comment that stages of recovery can affect the spasticity level. We have included Stroke patients in their chronic stage. Based on the current study’s inclusion/exclusion criteria, chronic Stroke patients who can maintain their balance in sitting and can move their upper limbs without external support were included. All of the patients’ modified Ashworth scale (MAS) scores ranged from 0 to 2. However, we did not assess their level of recovery through the Brunnstrom scale which is one of the limitations of this study.

[Comment 3]

What medications were patients taking?

Answer:

Many thanks for the reviewer’s comments. The patients were taking regular medicines to control their blood pressure and other medical symptoms like high blood cholesterol and diabetes if they had. All of the patient's medical conditions were stable and under control and they were recruited from the Medical Center of Hospital in Tainan, Taiwan by the rehabilitation medical doctor, and physical assessment to participate in our current study was conducted by an experienced physical therapist.

[Comment 4]

What was the level of fitness pre stroke?

Answer:

Many thanks for the reviewer’s comments. The pre-stroke fitness is difficult to assess because some of the patients had a stroke for many years and some of them have been for a few months. However, we recruited chronic stroke patients to test our device and record their range of motion by assessing their MAS score and relating both of these to predict the models for spasticity. However, in the future studies should include this variable before recruiting stroke patients.

[Comment 5]

Impact of smoking and recovery

Answer:

Many thanks for the reviewer’s comments. There would be the impact of smoking on the rate of recovery. However, we did not measure this parameter. Clinically after a stroke, medical doctors usually prohibit stroke patients to not smoking because it can affect their blood pressure and lead to recurrent incidents of stroke.

[Comment 6]

Presence or absence of partner influencing recovery?

Answer:

Many thanks for the reviewer’s comments. There would be an effect of psychological state on stroke recovery. Usually, stroke patients without a partner or family caretaker are presented with an altered psychological state and disturbed emotionally which can hinder their physical recovery in daily activities of motion. However, in our study, we did not assess this parameter. 

Reviewer 2 Report

Comments and Suggestions for Authors

The paper proposes a generalized estimating equation model to predict spasticity through measurement of upper limb active range of motion in stroke survivors. The study shows that AROM of upper limb joints can assist the subjective spasticity assessment (MAS) in stroke survivors.

Strenghts: The paper is well-written and coherently assembled. The research design is appropriate and results are properly analyzed.

Weakness: The control group was not age/gender matched with the subjects.

1. A hypothesis is alluded on line 88 but was not stated in the paper.

2. Figure 1 caption mentions joint angles, however, the figure defines segments and not the segment angles.

3. More details on the cone stacking test (line 154) will help. Also, Table 5 shows lower involvement in CS from upper limb joints, why? 

4. Please provide a list of variables tested (line 161) in the study. Was an attempt to build and test a reduced variable model made?

5. More details as to why quasi likelihood criteria (QIC) was selected (line  180) will be helpful. Some other studies, e.g., doi: 10.1111/j.2041-210X.2009.00009.x, have argued against using QIC.

6. Table 3 shows that the QIC in the case of right-dominant left-affected models were significantly lower than right-dominant right-affected despite no significant difference in the sample size, why so?

7. Fig. 4 shows a significant decrease in the ROM on the affected as well as unaffected side, why so?

8. Five subjects were excluded from the study, however, restuls in both Figure 4 and Table 6 appears to include them. 

9. The ROM comparison results for elbow and wrist (line 256) have very high p-values, why?

10. Please explain the statement on lines 186-187.

11. Please clarify the statement on lines 309-312.

Author Response

Author’s replies to the comments of Reviewer 2:

[Comment 1]

The paper proposes a generalized estimating equation model to predict spasticity through measurement of upper limb active range of motion in stroke survivors. The study shows that AROM of upper limb joints can assist the subjective spasticity assessment (MAS) in stroke survivors.

Strenghts: The paper is well-written and coherently assembled. The research design is appropriate and results are properly analyzed.

Weakness: The control group was not age/gender matched with the subjects.

Answer:

Many thanks for the reviewer’s comments. We agree with the comment that the control group was not age/gender matched. This is one of the limitations of our study which should be addressed in future research. We collected data a few years ago and the project is already closed and this limitation will be considered for future research.

[Comment 2]

A hypothesis is alluded on line 88 but was not stated in the paper.

Answer:

Many thanks for the reviewer’s comments. We agree with the comment and explained on lines 89-91 as:

Hence, to test our hypothesis for AROM and MAS score correlation for affected upper limb joints and solve the aforementioned limitation of using sensor data to assess spasticity in chronic stroke, we used a laboratory-developed wearable system and performed four voluntary tasks to record the AROM and assessed the spasticity level of the upper limb joints in stroke survivors by wearing a smart glove device on both upper limbs.

[Comment 3]

Figure 1 caption mentions joint angles, however, the figure defines segments and not the segment angles.

Answer:

Many thanks for the reviewer’s comments. We agree with the comment and revised figure 1 in the manuscript.

[Comment 4]

More details on the cone stacking test (line 154) will help. Also, Table 5 shows lower involvement in CS from upper limb joints, why?

Answer:

Many thanks for the reviewer’s comments. We agree with the comments and more details are added on lines 155-162:

For the CS task, two cone bases were arranged 20 cm from the edge of the table. Before the task, ten cones were placed near the side opposite to the participant’s test side, that is, if the left side were to be tested, the ten cones were placed on the cone base on the right side and vice-versa. The participants transferred the ten cones from one cone base to the other as rapidly as they could after getting the signal to begin the test. The participants were not allowed to lean their trunks to any side during the test. After moving all the cones, the participant returned the test hand to the initial position.

According to Table 5, the lower involvement of upper limb joints during the CS task may be due to the large joints involved like the elbow and wrist based on the nature of movement of the CS task. The task is to lift the cone from one place and put it into another place. Another reason may be that the affected side of hand functioning is not appropriate due to the weakness and spasticity of the upper limb in chronic stroke survivors.

[Comment 5]

Please provide a list of variables tested (line 161) in the study. Was an attempt to build and test a reduced variable model made?

Answer:

Many thanks for the reviewer’s comments. We agree with the comments. The list of the study variables as:

16 variables of upper limb joint angles as shown in Figure 1 and 4 variables of MAS score for elbow, wrist, thumb, and fingers.

However, we did not focus on the reduced variable model.

[Comment 6]

More details as to why quasi likelihood criteria (QIC) was selected (line  180) will be helpful. Some other studies, e.g., doi: 10.1111/j.2041-210X.2009.00009.x, have argued against using QIC.

Answer:

Many thanks for the reviewer’s comments. We agree with the comment but based on the above reference and other literature different models can be assessed by different parameters like: 1) the Akaike information criterion (AIC) using mixed models), 2) the quasi-information criterion (QIC) using generalized estimating equations, and 3) the deviance information criterion (DIC) using Bayesian models.

Barnett AG, Koper N, Dobson AJ, Schmiegelow F, Manseau M. Using information criteria to select the correct variance–covariance structure for longitudinal data in ecology. Methods in Ecology and Evolution. 2010 Mar;1(1):15-24.

According to Pan et al 2001, correlated response data are common in biomedical studies, and regression analysis based on generalized estimating equations (GEE) is an increasingly important method for such data. However, there seem to be few model selection criteria available in GEE. The well-known Akaike information criterion (AIC) cannot be directly applied since AIC is based on maximum likelihood estimation while GEE is non-likelihood based. So, a modification to AIC was proposed, where the likelihood is replaced by the quasi-likelihood and a proper adjustment is made for the penalty term. Hence, we have employed generalized estimating equations, and QIC is used to assess its efficacy for the correlation matrix.

Pan W. Akaike's information criterion in generalized estimating equations. Biometrics. 2001 Mar;57(1):120-5.

[Comment 7]

Table 3 shows that the QIC in the case of right-dominant left-affected models was significantly lower than right-dominant right-affected despite no significant difference in the sample size, why so?

Answer:

Many thanks for the reviewer’s comments. We agree with the comment. We have obtained a significantly lower QIC for the right-dominant left-affected models than right-dominant right-affected models which is due to the random effect of data and a little difference in the sample size, or due to better ability to perform AROM activities on the left-affected side than the right-affected side.

[Comment 8]

Fig. 4 shows a significant decrease in the ROM on the affected as well as unaffected side, why so?

Answer:

Many thanks for the reviewer’s comments. We agree with the comment. This comparison was made with healthy controls’ ROM and the decrease in ROM on both sides may be due to weakness after stroke and the aging effect in unaffected and affected sides. But if we compare stroke survivors’ ROM then the unaffected side ROM is higher than the affected side. 

[Comment 9]

Five subjects were excluded from the study, however, results in both Figure 4 and Table 6 appears to include them.

Answer:

Many thanks for the reviewer’s comments. We agree with the comment and out of 41 stroke subjects, five were not included in GEE model computation. But in Figure 4 and Table 6, we have presented all of the subjects' ROM data. The reason for not including five subjects was due to a very low number of subjects and the GEE model cannot run.    

[Comment 10]

The ROM comparison results for elbow and wrist (line 256) have very high p-values, why?

Answer:

Many thanks for the reviewer’s comments. We agree with the comment. The ROM for the affected and unaffected sides of the elbow and the wrist joints changed but did not achieve significant (lower) p-values and had higher p-values. The reason could be due to the lack of higher difference in muscle strength in both affected and unaffected sides as all of the patients' MAS score is in the range of mild to moderate spasticity and they can move their upper limbs. However, this trend was not observed in finger ROMs because of complex joint involvement.

[Comment 11]

Please explain the statement on lines 186-187.

Answer:

Many thanks for the reviewer’s comments. It explains the gender vise distribution of healthy control and stroke groups and is revised for clarity on lines 192-194.

[Comment 12]

Please clarify the statement on lines 309-312.

Answer:

Many thanks for the reviewer’s comments. We did not measure the tonic strength reflex threshold (TSRT) which is important to determine the objective spasticity. Also, we did not measure the sEMG in our study which can be measured in future studies and included for GEE model estimation based on the proposed approach. We also did not monitor the speed of the voluntary tasks which should be important to determine speed-specific GEE models for spasticity assessment.
